# Cardiovascular Disease in Women’s Prisons: A Qualitative Study of Dietary Habits from the Perspective of Professionals

**DOI:** 10.3390/nu17091428

**Published:** 2025-04-24

**Authors:** Ana Margarida Machado, Iara Rafaela Ferreira, Mariana Rodrigues, Adriana Taveira, Francisca Linhares, Ana Paula Macedo

**Affiliations:** 1Health Sciences Research Unit: Nursing (UICISA: E), Coimbra School of Nursing, 3046-851 Coimbra, Portugal; anamargaridaferreiramachado@gmail.com (A.M.M.); b14265@ese.uminho.pt (I.R.F.); adricasofia@gmail.com (A.T.); 2Postgraduate Programme in Nursing (PPGENF), Federal University of Pernambuco (UFPE), Recife 50670-901, Brazil; mariana.acioly@ufpe.br (M.R.); francisca.linhares@ufpe.br (F.L.); 3Higher School of Health, University of Trás-os-Montes and Alto Douro (UTAD), 5000-801 Vila Real, Portugal

**Keywords:** prison, women’s health, eating habits, cardiovascular diseases, social vulnerability

## Abstract

**Background**: Cardiovascular disease (CVD) is the leading cause of death worldwide and is exacerbated by poor dietary habits, particularly in settings such as women’s prisons. Incarcerated women are often exposed to ultra-processed foods, limited nutritional education, and restricted living conditions that increase their risk of CVD. **Objectives:** This study aimed to explore the challenges perceived by professionals in a Portuguese women’s prison regarding the prevention of CVD, particularly through dietary interventions. **Methods**: A qualitative, exploratory and descriptive study was conducted using a focus group with six professionals. Data were collected in July 2024 and analysed using thematic content analysis. **Results**: Three thematic categories emerged: (1) contextual challenges of the prison system (e.g., sedentary lifestyle, limited food options); (2) socio-cultural resistance to behavioural change (e.g., low adherence to health programmes, use of food as a coping mechanism); and (3) the need for sustainable and interdisciplinary intervention strategies. **Conclusions**: The findings highlight the complexity of promoting cardiovascular health in female prisoners. Interventions need to take into account mental health support, prisoner autonomy and institutional constraints. Future research should develop and test targeted, context-specific nutrition programmes in similar settings.

## 1. Introduction

Cardiovascular disease (CVD) is the leading cause of morbidity and mortality worldwide, accounting for approximately 17.9 million deaths annually, or 32% of all deaths worldwide [1]. The World Health Organisation highlights that most CVD deaths are preventable through early detection and management of behavioural risk factors, such as unhealthy diet, physical inactivity and tobacco use [1,2]. In Portugal, CVDs accounted for 27% of all deaths in 2020, highlighting their national importance [3].

These problems are particularly acute in prisons, where environmental and structural conditions exacerbate health inequalities. Prisoners experience a disproportionate burden of chronic disease due to socio-economic vulnerability, limited access to health care, overcrowding, psychological stress and limited opportunities for physical activity [4,5]. While all prisoners are exposed to these conditions, women are particularly vulnerable due to both biological and gendered social determinants of health [4,6,7]. Despite this, women in prison remain underrepresented in research, particularly concerning cardiovascular health.

The novelty of the present study lies in its specific focus on the intersection between dietary habits and cardiovascular risk in women’s prisons, an area that has received limited attention in the existing literature. Although some studies acknowledge the relationship between incarceration and poor diet [8,9,10], few examine how these factors are perceived by professionals working directly with incarcerated women [11,12]. Understanding their perspectives may reveal institutional and behavioural barriers to effective CVD prevention strategies.

Prison diets are often inadequate due to budget constraints, poor supervision and a tendency to prioritise cost over quality [8]. This results in meals high in saturated fat, sugar and sodium and low in fresh produce and fibre. These dietary deficiencies contribute to a higher prevalence of obesity, hypertension, diabetes and metabolic syndrome—conditions strongly associated with cardiovascular disease [9]. Studies have shown that poor diet not only increases physiological risk but also exacerbates existing health inequalities in the prison population, particularly among women [9,10,13]. In women’s prisons, dietary behaviours are further complicated by emotional distress, the use of food as a coping mechanism, and limited autonomy over food choices. Commissary options are often dominated by ultra-processed foods, contributing to a pattern of poor diet that significantly increases cardiovascular risk [10,13].

Despite growing awareness of these challenges, there is a notable lack of empirical research exploring how professionals working in women’s prisons perceive and address the nutritional components of cardiovascular risk. This study aims to bridge this gap by exploring professionals’ perspectives on the challenges and strategies of promoting cardiovascular health through nutritional interventions in a Portuguese women’s prison. By capturing their lived experiences, the study provides valuable insights into the systemic barriers and opportunities for intervention, contributing to the development of evidence-based and contextually appropriate health promotion strategies in correctional settings.

## 2. Materials and Methods

### 2.1. Study Design

This was a qualitative, exploratory and descriptive study conducted at a female prison facility in Portugal in July 2024. This study design was chosen to enable an in-depth exploration of the distinctive challenges faced in the prevention of CVDs related to eating habits within this specific context. The qualitative approach was deemed the most suitable to capture the complexity of the socio-cultural and contextual factors influencing cardiovascular health and eating habits in this vulnerable population.

The use of focus groups as the primary data collection method was guided by the need to encourage interaction and discussion among participants, which is a valuable strategy for uncovering shared experiences and institutional dynamics [14]. This method allows for the emergence of shared meanings and divergent perspectives on sensitive and complex issues and is widely recommended in qualitative health research for its ability to generate rich, nuanced data that reflect collective and individual viewpoints.

The focus group guide was developed based on a literature review, expert consultation, and discussions among the research team and covered topics such as perceived barriers and facilitators to healthy eating, and institutional food services and policies. This guide was pilot tested with a professional outside the study site to ensure clarity and relevance.

### 2.2. Participants

A non-probabilistic purposive sampling technique was employed, with participants selected based on their interest in the study and availability, ensuring that institutional functioning was not disrupted. This choice is justified by the exploratory nature of the research and the operational constraints of working in a closed institutional environment, where voluntary participation and logistical feasibility significantly shape recruitment.

Inclusion criteria comprised (a) professionals currently employed at the prison facility with a range of functional roles involving direct interaction with incarcerated women in order to guarantee the representativeness of all professional categories, such as healthcare providers (e.g., nurses, psychologists, physicians), social workers and security personnel, regardless of their academic background, and (b) a minimum of six months of experience at the facility to ensure contextual familiarity. The principle of homogeneity among the group of participants [14] is achieved by the creation of different focus groups according to professional category. Each focus group will respect the theoretical limit of 6 to 10 participants and the number of sessions will be determined by reaching theoretical data saturation, supporting the adequacy of the sample and successfully fulfilling the outlined research objectives [11,14,15].

It is decided to hold the meetings on-site at the prison, in a private room designated for this purpose and, to support participation, we considered several time slots to minimise disruption to daily operations.

### 2.3. Data Collection

Study participants were provided with a disclosure page outlining the study’s purpose and the intent for its subsequent publication. A comprehensive focus group guide was used to ensure consistency, and the discussion lasted approximately ninety minutes.

The session was audio-recorded using a digital device, with prior informed consent from participants, to ensure accurate transcription. The resulting data were transcribed verbatim, but no direct identifiers were collected, and strict confidentiality measures were maintained throughout the study to ensure participant privacy and confidentiality. In addition, participants completed a socio-demographic questionnaire, collecting data on age, gender, marital status, area of residence, education, professional training, role within the prison, and length of professional experience in that setting.

The session was moderated by a researcher experienced in qualitative methods, supported by two other researchers—one responsible for recording facial expressions, and another taking summary notes. Each participant had an equal opportunity to contribute.

### 2.4. Ethical Considerations

Ethical approval for the study was granted by Ethics Subcommittee for Life and Health Sciences at the University of Minho (Approval number CEICVS 073/2024, Approval date 26 June 2024) and the Directorate-General for Reintegration and Prison Services (DGRPS) authorised it for realisation (Approval number DGRSP 96/CCCRE, Approval date 28 June 2024).

### 2.5. Data Analysis

Thematic content analysis was conducted following Bardin’s methodological framework [11]. Statements from participants were transcribed and identified with the letter P (Professional), followed by a numerical sequence indicating the order in which they first spoke.

Analysis was conducted manually, relying on a structured and systematic manual approach, without the use of qualitative data analysis specialised software.

A mixed deductive and inductive coding approach was applied. Initially, broad categories based on the existing literature regarding CVD prevention and prison health were pre-defined, deductively guiding the first cycle of coding. Subsequently, as the transcripts were reviewed, new codes particularly regarding specific challenges and practices related to eating habits were derived inductively from the participants’ discourse. These inductively derived codes were then grouped under or contrasted with the pre-defined categories, enabling a dynamic interplay between theoretical constructs and data-driven insights.

Three independent researchers compared and discussed their coding, reaching a consensus on the codes that most accurately captured participants’ perspectives. In the second round, two further refined the thematic structure for consistency. Similar codes were grouped into broader categories, and relationships between these categories were explored.

Following Bardin’s approach [11], the main themes were operationalised by analysing sentences as context units within participants’ statements, as shown in Table 1. This method enabled a deeper understanding of the conditions in which these perspectives emerged, shedding light on the institutional policies and structural factors influencing their experiences.

## 3. Results

### 3.1. Participant Socio-Demographics

The sample consisted of six healthcare professionals who worked in the prison facility that served as the study setting, with no participation of the inmates. A single session was conducted on-site at the prison, in a private room designated for this purpose. To support participation, several time slots were initially considered, and the final schedule was agreed upon collaboratively to minimise disruption to daily operations.

Four of these professionals were married, and two were single. In terms of gender, four were male and two were female. The average age of the professionals was 53.67 (±9.651), with ages ranging from 47 to 74 years old. All professionals lived in the surrounding metropolitan area. Although they are all health professionals, the six individuals have been working in different fields in the context for approximately 20 years, including coordination, clinical direction and management, training, psychology and nursing.

It is worth noting that not all clinical service professionals were able to participate in the study due to the need to maintain security and ensure the quality of care that was taking place in the facility.

### 3.2. Categories

Table 2 presents the final categories with their representative quotes. Three categories were extracted deductively from the data collected. These included (1) contextual challenges of the prison system; (2) socio-cultural resistance to behaviour change, and (3) the need for sustainable and interdisciplinary intervention strategies. Each category provided a structured approach to gather valuable insights into the diagnosing of concrete needs, promoting reflection and developing new strategies for implementation.

#### 3.2.1. Contextual Challenges of the Prison System

In this category the professionals showed knowledge and endeavour in nutritional assessment and primary and secondary prevention of cardiovascular diseases. However, they realised that the success of these interventions faces many challenges inherent in the contextual nature of the prison system.

The opinions of all professionals regarding the identified challenges are convergent and indicate, for example, that sedentarism and limited dietary diversity are linked to the conditions imposed in prison. According to the professionals, a more contextually appropriate health education approach is needed, aligned with the educational level of the inmates, who often prioritise other needs beyond food care. Other challenges are also highlighted by professionals, such as smoking, which is considered a means of relieving tension and stress within the prison environment. Clinical service professionals report to providing treatment in more complex situations to manage recurring self-harm behaviours, such as self-mutilation and para-suicide.

These challenges are reenforced and demonstrated by the following quotes (Table 2: Category 1—Quotes P1 to P6).

In response to the various contextual challenges posed by the prison system, the participants’ facial expressions conveyed agreement as they nodded affirmatively to each other’s interventions. When discussing smoking cessation, the professionals laughed and ridiculed the topic, perceiving it as a utopian idea, yet also as a sentiment of empathy.

In the context of cardiovascular disease prevention measures, these findings are concerning, as they highlight that significant risk factors (sedentarism, unhealthy diet, and smoking) for cardiovascular diseases are difficult to control within the prison environment of the study.

#### 3.2.2. Socio-Cultural Resistance to Behaviour Change

This category captures the strong resistance to behavioural change by inmates from the perspective of professionals, along with the need to prevent riots and conflict among inmates had led to many attempts to failing.

According to professionals, resistance to behavioural change is linked to socio-cultural factors such as the lifestyles and customs of the inmates, as well as the presence of a small market where inmates are free to make their own choices, whether harmful to their health or not. This is in line with the current legislation, which states that every incarcerated individual retains all the rights of any other citizen, except those restricted as a direct consequence of imprisonment.

These factors are reinforced by the following quotes (Table 2: Category 2—Quotes P2, P3 and P6).

It is worth mentioning that, faced with socio-cultural resistance to behavioural change, the participants’ facial expressions conveyed agreement.

#### 3.2.3. The Need for Sustainable and Interdisciplinary Intervention Strategies

This category elucidates healthy eating as a sensitive but urgent topic that required attention. Although various efforts had been made, progress remained slow and the journey ahead is long.

The professionals consider that there are many processed foods within the prison context and that awareness in the dining hall and restrictions on food purchases are necessary in prolonged situations, alongside interdisciplinary support and state legislation. The professionals also report that, for example, there is a reduction in salt in the food, but it is a futile effort since there is no control over what inmates can buy in the small market. Furthermore, they point out that psychiatric treatment itself leads to demotivation, making it equally necessary for inmates to take responsibility for their own health.

This intervention strategies are pointed by the following quotes (Table 2: Category 3—Quotes P2, P3, P5 and P6).

Thus, the need to avoid unrest, the presence of the small market, and unrestricted access to industrialised products sold in the establishment, supported by current legislation, weaken food control measures for the prevention of cardiovascular disease.

Faced with the need for sustainable and interdisciplinary intervention strategies, the participants’ facial expressions conveyed agreement. When discussing state legislation, one participant was observed to gesture constantly with their hands, while all others nodded affirmatively, demonstrating both dissatisfaction and an interest in intervening in the situation. The question took them by surprise; it was manifestly unexpected, leaving all individuals feeling quite uncomfortable.

The discomfort expressed by the participants is associated with a sense of powerlessness in relation to the implementation of control measures. This is largely attributed to the perceived lack of state support and the absence of effective legislation to improve conditions in Portuguese prisons. This perception was further reinforced by objections raised to one of the proposals put forward by a researcher, which advocated for the implementation of interdisciplinary interventions within the prison environment. Specifically, this proposal entailed the utilisation of non-food reward systems for inmates.

The professionals expressed reservations about the viability of such rewards, contending that their effectiveness would be contingent upon their implementation within a rigidly controlled institutional environment. Furthermore, emphasis was placed on the contemporary reality that the prospect of external financial assistance, specifically from family members, endows inmates with a degree of purchasing power that undermines the efficacy of internal strategies.

In this sense, the participants proposed that any intervention should promote the active participation of the inmates themselves, as well as the strategic involvement of the media, considering the high amount of time inmates are exposed to television and radio and the potential of this medium for organising activities with educational and reintegration objectives.

## 4. Discussion

The results presented transparently reflect the sharing by professionals in this female prison context of the contextual, socio-cultural and structural challenges they have experienced about the difficulties faced in preventing CVD in the establishment, with an emphasis on the eating habits of the women prisoners.

### 4.1. Contextual Challenges

Professionals working in the prison system recognise the importance of nutritional assessment and the implementation of both primary and secondary prevention strategies for cardiovascular disease. However, their ability to enact meaningful change is severely constrained by structural and environmental challenges inherent to incarceration. One of the most significant barriers is the forced sedentary lifestyle, which limits opportunities for physical activity and exacerbates health risks as proven by other studies already carried out [5,16]. While some facilities offer structured exercise programmes such as gym classes, yoga, and team sports, professionals report low motivation and participation among inmates, reflecting broader challenges in promoting long-term behavioural changes [17]. Studies carried out in different regions corroborate these findings. An integrative review identified that the most frequent risk factors for cardiovascular diseases among people deprived of their liberty include overweight, obesity, hypertension, dyslipidaemia, diabetes, smoking, excessive alcohol and drug consumption, sedentary lifestyle, metabolic syndrome, anxiety, depression and unhealthy diets. These factors are potentiated by an environment with few opportunities to change these habits [4]. In addition, a cross-sectional study carried out in a male penitentiary revealed alarming prevalences of hypertension (24.8%), dyslipidaemia (54.5%), overweight (49.9%), metabolic syndrome (16.8%) and diabetes (2.5%). Difficulty in accessing health services, combined with long sentences and an unhealthy environment, encourages the development and worsening of chronic diseases, representing a significant challenge for the organisation of prison health care [18].

The quality and variety of food available in prison settings further complicates health interventions. In many institutions, meals are outsourced to external contractors, resulting in nutritionally inadequate food that often fails to meet dietary recommendations [8]. A lack of fresh fruits and vegetables, combined with the over-reliance on processed foods, contributes to poor diet quality, increasing the risk of obesity, hypertension, and other cardiovascular conditions [9]. These dietary limitations not only hinder CVD prevention efforts but also reduce adherence to broader health strategies aimed at improving inmate well-being [13]. In Brazil, a study assessed the menu and physical–structural conditions of food and nutrition units in prisons, highlighting that offering poor quality food can be seen as a form of oppression and dehumanisation. The research emphasised that prison food must be adequate, balanced and varied, as established by national and international legislation, in order to guarantee the dignity and health of prisoners [19]. A recent study analysed the eating experiences of individuals in prisons globally, showing that food in this type of institution is often characterised by insufficient portions, low nutritional quality and limited dietary diversity—factors that reflect the structural and organisational constraints of the prison system. The lack of choice and control over food was identified as an element that contributes significantly to the loss of individual autonomy, with negative repercussions on prisoners’ physical health, psychological well-being and perception of dignity [20]. In contrast, this study found that women prisoners had a considerable degree of autonomy in terms of food, provided by the existence of an in-house supermarket. However, this autonomy did not translate into better results in terms of food quality, suggesting that simply providing choice is not enough on its own to promote healthier eating practices in this context.

The low level of education and health literacy among incarcerated women is also a significant challenge for the adoption of healthy eating behaviours and the prevention of chronic diseases, such as cardiovascular diseases. A large proportion of prisoners often have a limited understanding of the long-term benefits of healthy eating, which hampers adherence to dietary interventions [12]. According to a study carried out in Portugal, this lack of literacy is exacerbated by pre-existing socio-economic and educational factors [21].

In this study, professionals report that although health interventions are a priority, immediate concerns, such as smoking cessation and prevention of self-injurious behaviour, take precedence over nutrition, a fact corroborated by several studies [22,23]. The competing demands on prison healthcare services, combined with limited resources, make it difficult to implement sustainable dietary improvements.

### 4.2. Socio-Cultural Barriers

The perspectives of the professionals interviewed show that socio-cultural factors, such as low levels of education and health literacy, unfavourable socio-economic conditions and family fragility, are significant barriers to the promotion of healthy eating behaviours in prisons.

As mentioned above, the lack of education and health literacy significantly jeopardises understanding of the implications associated with an inadequate diet [21]. At the same time, many of the women in prison come from socio-economic backgrounds characterised by food insecurity, unhealthy eating patterns and limited access to health care [24], factors which contribute to the consolidation of dysfunctional eating behaviours during the period of imprisonment.

During the focus group, it was observed that responsibility for health is often attributed to prison staff, reflecting a passive attitude on the part of inmates [12]. This perception makes it difficult to promote autonomy and individual responsibility in the prevention of cardiovascular diseases [8]. The policy of ‘freedom within walls’, although it promotes personal food choices, makes it difficult to apply nutritional strategies orientated towards public health [9].

Initiatives aimed at improving the nutritional profile of the products available in the canteen are often met with resistance, as any restrictions on foods considered ‘favourite’ are interpreted as infringing on individual rights. This conflict between promoting institutional health and preserving personal autonomy raises important ethical and operational questions. In fact, limiting access to processed foods can trigger significant adverse reactions, such as hunger strikes, bartering practices between inmates and, in extreme cases, self-injurious behaviour [13].

In addition, family breakdown and the absence of support networks are relevant factors in weakening the emotional state of individuals deprived of their liberty, a context in which eating often takes on a compensatory or secondary role in relation to other needs perceived as more urgent. Empirical evidence indicates that psychological stress is an inherent condition of imprisonment and has a significant impact on the formation of eating behaviours. Studies also show a consistent association between chronic stress and the development and maintenance of eating disorders such as anorexia, bulimia and binge eating. Such disorders not only make it difficult to adopt healthy eating habits but they can also worsen inmates’ general clinical condition [25].

In addition, prolonged emotional stress often leads to the use of dysfunctional coping strategies, such as smoking and the consumption of ultra-processed foods—behaviours that are widely recognised in the scientific literature as risk factors for cardiovascular disease [23]. In this sense, as long as the psychosocial determinants underlying maladaptive eating behaviours are not adequately addressed, efforts to promote healthier eating and mitigate cardiovascular risks will remain limited and insufficient.

### 4.3. Intervention Strategies

The implementation of sustainable health interventions in prison settings faces significant structural and cultural obstacles, requiring a multidisciplinary approach that balances inmates’ resistance with the administrative restrictions inherent in the correctional system.

One of the measures highlighted by the participants was reducing the salt content of the main meals (lunch and dinner). However, it was also pointed out that although this initiative represents a positive step forward, it ends up having limited effectiveness due to the autonomy granted to inmates in purchasing ultra-processed products—such as industrialised sauces (mayonnaise, ketchup) and snacks (packet crisps)—available in the in-house mini-market. If there are to be significant changes in eating habits, a reformulation of the institutional paradigm is essential, with the development of more comprehensive food policies that regulate both supply and food choices, while preserving the autonomy of inmates [8].

These strategies should be designed and implemented taking mental health into account as a central factor, given its decisive role in the success of any intervention [12]. Without adequate psychological support, attempts to change behaviours, whether they are eating or smoking-related, are unlikely to generate lasting effects in promoting cardiovascular health in a prison environment [23].

The professionals surveyed also emphasise that the effectiveness of interventions depends on active collaboration between all the actors in the prison system, going beyond the scope of health professionals and including the security team, administrators and the prisoners themselves. The success of health promotion initiatives, therefore, requires a coordinated and sustained effort, supported by clear, coherent and applicable public policies that take into account the particularities of the prison environment [5]. In the absence of institutional commitment at multiple levels, interventions tend to remain fragmented, making them difficult to sustain and limiting their long-term impact on the health of the prison population.

Several studies corroborate the conclusions about the importance of interventions to improve mental health and the quality of life of prisoners. One study evaluated the effectiveness of mental health promotion programmes, highlighting significant improvements in self-esteem and a reduction in anxiety among participants [26]. Another study investigated the effects of the ‘Generating Social Pathways’ programmes on the psychosocial rehabilitation of prisoners, showing substantial reductions in cognitive distortions and dysfunctional beliefs associated with antisocial behaviour [27]. In this context, a study carried out in Brazil proposes various strategies to improve food and health policies in the prison system, whose suggestions also apply to international contexts. Among the main strategies are: (1) integration of inter-institutional public policies; (2) nutritional adequacy of meals; (3) training and capacity building of professionals; (4) promotion of inmate participation; (5) adoption of food safety policies; and (6) attention to psycho-social conditions [28]. By implementing a strategic plan as mentioned, correctional facilities have the potential to create a healthier and more rehabilitative environment, effectively addressing the complex needs of the incarcerated population.

In light of the professionals’ expressed feelings of discomfort (see Table 2), this can be substantiated by the absence of efficacious strategies and standards for the management of inmates. This observation underscores the necessity for a comprehensive review of prevailing practices and legal frameworks [29]. It is reasonable to hypothesise that the legal obligation and current health guidelines specific to the prison context will result in an improvement in prison environments and the health of inmates.

In the context of these practices, it is necessary to understand which rewards are most appropriate to encourage inmates to maintain good behaviour. It was established through previous research that food rewards are not an appropriate incentive, given their potential interference with mental health and the possibility of triggering eating disorders [30]. In this sense, a study carried out in a prison setting identified various types of incentives used to promote desirable behaviour among inmates. Of these, tangible rewards were identified as being of particular significance, including the granting of additional privileges or access to recreational activities. One of the researchers suggested that cinema sessions could be used as an incentive [31].

In this regard, the participants proposed a collaborative endeavour with the media, encompassing both television and radio platforms. A preceding study has highlighted the potential of the media to function as an educational and awareness-raising instrument. The efficacy of health education initiatives in increasing knowledge about pathologies and promoting healthy lifestyle habits, such as a balanced diet, was demonstrated [32]. These initiatives are most effective when transmitted via these specified means.

### 4.4. Institutional Opportunities and Constraints for Health Promotion

This study was conducted in a prison that is noteworthy on a national scale for its exceptional conditions. The institution boasts cutting-edge infrastructure, and a plethora of resources and activities designed to foster optimal physical and mental well-being. These include yoga and gymnastics classes, a well-equipped gym, sports activities, and a comprehensive range of hairdressing, laundry, and childcare services. The context is conducive to the implementation of health promotion strategies and the adoption of healthier lifestyles, conditions that are less prevalent in other prisons. Consequently, the institution’s preferred environment is conducive to the development of more effective and sustainable interventions.

Nevertheless, a significant limitation in the approach to smoking cessation on the part of the institution’s professionals, many of whom are also smokers, was identified. The programme participants were observed to demonstrate a capacity for empathy, which enables them to comprehend the factors that precipitate elevated levels of tobacco consumption among the prison population. This ability to empathise is concomitant with a sense of identification with the plight of the inmates. This perception of similarity has the potential to engender a less proactive stance with regard to the development and implementation of smoking cessation strategies, particularly within a prison environment characterised by elevated levels of stress and anxiety. In this manner, despite endeavours to encourage health, the individual actions of professionals have the potential to act as an impediment.

Furthermore, adherence to and maintenance of healthy eating practices is limited. The presence of a minimarket, offering unrestricted access to a wide range of food products, including less healthy options, hinders the pursuit of a balanced diet. Moreover, within the context of economic constraints, inmates may encounter difficulties in procuring desired products, which can result in informal exchanges among their peers. It is important to note that the profits generated by this minimarket are reinvested in benefits for the prison population, such as the acquisition of corrective eyewear.

### 4.5. Strengths and Limitations

This study presents a solid methodological approach that is coherent with its objectives, namely using a qualitative, exploratory and descriptive design, which is appropriate for analysing professional perspectives in the prison context. The use of a focus group, supported by a validated script and enriched with non-verbal observations and field notes, allowed for a rich and contextualised data collection process. Given that the group was only made up of experienced professionals from the institution, there was a coherent discourse, and an in-depth exploration of the topics discussed.

Nevertheless, this study has some important limitations, namely the small sample size, the fact that it was carried out in a women’s prison and the absence of inmates’ perspectives, reducing the potential for data triangulation. In addition, the prison context in which the research took place—characterised by infrastructure and conditions above the national average—may limit the generalisability of the results to other institutions with less favourable conditions.

It is important to emphasise the scarcity of studies in the scientific literature exploring the eating habits of women in prison and their relationship with the incidence and worsening of cardiovascular diseases in this population.

## 5. Conclusions

This study highlights the challenges of promoting cardiovascular health in women’s prisons, with a particular focus on inmates’ eating habits. Structural and socio-cultural obstacles were identified, such as the imposed sedentary lifestyle, the limited food on offer, low levels of health literacy and the behavioural resistance of inmates. Although the existence of a mini market within the prison promotes autonomy, it encourages the consumption of ultra-processed products, jeopardising healthy eating strategies. In addition, the impact of stress and deprivation of liberty encourages compensatory behaviours, such as smoking and excessive intake of sugars and fats.

In this sense, it is essential to adopt an interdisciplinary and integrated approach, involving health teams, prison management and inmates themselves in the development of effective strategies. The implementation of eating education programmes, combined with the strengthening of mental health, can help to reduce the risks identified. The results of this study underscore the need for more effective and humanised public policies capable of improving prison food conditions and, consequently, the quality of life of inmates.

Specifically, the results presented may imply a revision of dietary guidelines in prisons and adjustments to the menus offered, the development of interdisciplinary educational programmes that include schoolteachers and take into account the educational levels of the inmates, the provision of continuous education for professionals working in this context, and the development of strategies to promote inmate autonomy, such as the cultivation of community gardens within the prison facility.

It is recommended that further studies be conducted to identify more comprehensive policy initiatives aimed at improving nutritional conditions and promoting interdisciplinary actions for the prevention and mitigation of cardiovascular diseases in women’s prisons.

## Figures and Tables

**Table 1 nutrients-17-01428-t001:** Diagram representing Bardin’s assumptions (adapted from [11]).

Content Analysis Operations	Cardiovascular Diseases in Women’s Prisons: Eating Habits from the Perspective of Professionals		
Categories	Contextual challenges of the prison system	Socio-cultural resistance to behavioural change	The need for sustainable and interdisciplinary intervention strategies
Constitution of the analytical corpus	The material relating to the participants’ statements was obtained through voice recording. A socio-demographic questionnaire was used to characterise the participants. The observer recorded the participants’ facial expressions in detail and the note-taker summarised the topics of discussion during the session.		
Context units	Formal registration unit: a minimal fragment of content, such as a sentence or a statement, taken from the analytical corpus was used to identify or characterise each category.		

**Table 2 nutrients-17-01428-t002:** Key categories.

Categories	Quotes
Contextual challenges of the prison system	P3—(…) prisoners (…) spend many hours in their cells (…) it’s a cubicle, they take two steps and get from one side to the other, a sedentary lifestyle is compulsory, it’s impossible for it not to have an impact on health. […] P5—(…) we have to contextualise the environment we have, because: we’re in the “outside population” and we know that a sedentary lifestyle is a high risk for cardiovascular diseases, but here… we try to raise awareness (…) sometimes health education in prisons isn’t about the strategies and the approach (…) we have to contextualise the environment we’re in (…) I can’t say to a woman, “Eat a variety of fruit”, when it’s essentially three pieces of fruit served. (…) So we have to direct, (…) realise (…) the environment we have and the resources we have, on all sides, to make this approach, which is sometimes a contradiction to what we have out there… […] P2—And our big obstacle in this environment is the education of the inmates themselves, their level of schooling is very low (…), and they have much more important priorities than not putting salt in their food and giving up smoking (…). […] P6—(…) Within prisons, we have something vital, which is smoking. (…) in these people who are in a confined space from 7 p.m. until 7 a.m., at least 12 h in a given space (…). […] P4—We’ve even had some smoking cessation programmes, with strong communication. Still, the effectiveness is… (…) it’s a context of a lot of tension, a lot of stress, (…) every day a person is subjected to enormous tension (…). […] P5—Even we professionals, watch out! Even we professionals! (…). […] P4—(…) in terms of the work of all the clinical services, we do a lot of treatment in slightly more complicated situations (…) both in terms of smoking reduction… and often in terms of behaviour. And sometimes, unfortunately, the reward itself is the cigarette. Because the system is like that (…) because our capacity to act is also very limited. […] P1—We have a recurrence of self-injurious behaviour (…). […] P4—(…) We’re talking about very complicated personalities. […] P6—(…) the phenomenon of self-mutilation, (…) para-suicide. […]
2. Socio-cultural resistance to behavioural change	P3—(…) Well, we have a mini market here, (…) just like (…) outside, (…) it sells everything… and (…) You can’t forbid people from having their own choices, but you have to see if those choices (…) could be extremely harmful to their health (…). […] P6—(…) The positive nuance (…) of this minimarket is that the profits from the “canteen” (…) always go to the inmates themselves. (…) The downside, without a doubt, is that they have all kinds of products at their disposal (…) and they have to, as part of their accountability process, realise what is good for them and what isn’t. […] P2—(…) If they have financial support from outside, they don’t even need to go to work, and so, on the question of the user’s responsibility, if there was a change in mentality, it would be revolutionary (…)! […] P3—(…) If our legislation says straight out, in black and white, that the citizen prisoner has all the rights as any other citizen, except those resulting from the prison itself (…) it’s all designed to (…) relieve the prisoner of responsibility! […] P6—(…) The system wants there to be no riots, no rebellions, and for people to be relatively calm. This is sometimes counterproductive to what we’re working on (…). […]
3. The need for sustainable and interdisciplinary intervention strategies	P5—[We think some care should be taken when approaching] The food part, even the products, everything that’s on sale in the canteen, and a lot of processed foods that exist (…) Even (…) sensitising the canteen, and (…) using measures to restrict the purchase of some foods in extended situations… (…) Maybe today we need to eat a chocolate, or an ice cream, or a cake, whatever. Everyone does. Now, it can’t be that the meal is replaced by whatever, can it? (…) There’s a reduction. For example, in the kitchen, (…) there’s a reduction in salt in the food, but then there’s no control, and they can buy ketchup, mustard, and mayonnaise in the canteen. The food comes with reduced salt, but then there’s a bottle of ketchup and mustard in your pocket. (…) Maybe if the guards were more sensitive…, but they can only warn, they can’t take it away from them (…). […] P5—The evolution is positive, they’re much more sensitised, they’re more careful, (…) but then there are also other situations here (…) … the psychiatric part, even in terms of medication, which often (…) also leads to demotivation (…). the psychiatric part, even in terms of medication, which often (…) also leads to excess weight (…) there’s a lack of motivation (…). […] P2—(…) Things have changed, it’s true, but so has the profile of the inmate (…). […] P3—But (…) our legislation (…). […] P6—(…) The effort must be multidisciplinary, concentrated, and… resources must be maximised. […] P3—But before that, and this is the hardest part, it’s up to each state, each country, to create the necessary conditions so that all this can be implemented. […] P2—The issue of inmates taking responsibility (…) is very important! One of the things we’ve noticed for many years (…) is that they almost take no responsibility for everything to do with their health and think it’s our obligation, one way, to resolve all situations, whatever they do. […]

## Data Availability

The data supporting this study are not publicly available due to ethical and privacy restrictions related to the confidentiality of participants and the sensitive nature of the prison environment. Access to the data is restricted in compliance with ethical guidelines and institutional policies.

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
