# Peer review of "Cardiovascular Disease in Women’s Prisons: A Qualitative Study of Dietary Habits from the Perspective of Professionals"

_nutrients, 2025, doi:10.3390/nu17091428_

Round 1

Reviewer 1 Report

Comments and Suggestions for Authors

It would be good to include the type of research in the title.
The introduction sufficiently establishes the need for the research. It is positive that qualitative research was conducted, such research is rare these days. The methodology is described in detail and is easy to follow.
The authors also explained the limitations of the research in detail, and they did a wide-ranging and thorough job when writing the article. The topic itself does not fundamentally affect large populations, so its public health significance is moderate, but it draws attention to an important issue. Many generalities are also revealed, which could have been suspected with great certainty even without the research (for example, lack of motivation, lack of health literacy). Perhaps the greatest value of the article is that, according to their own admission, the survey was conducted in an outstandingly equipped prison and even then they revealed major shortcomings.

Author Response

Thank you for your thoughtful feedback and for highlighting the positive aspects of our work. We appreciate your recognition of the study’s value, particularly in the context of the well-equipped prison setting. We have taken your suggestion regarding the title into consideration and made the appropriate changes, as you can read in the manuscript.

Reviewer 2 Report

Comments and Suggestions for Authors

The manuscript by Machado et al reports the barriers professionals meet in the prevention of CVD in a women’s prison, with focus on eating habits. The perspective of the professionals is important, but the manuscript has some limitations and needs to be revised. Please review the following  comments:

  1. Remove the reference in the abstract
  2. The results section in the abstract is very short, and it seems that some of the findings are presented in the background section? Consider revising.
  3. A literature-reasoning behind the sampling technique and chosen method (focus groups and interview form) should be included in Methods. Also, the development of the interview guide should be specified. “The methodological procedure resulted from…” does not give enough information.
  4. In 2.2, more description of the session specifics ahould be added. Were several time points offered to facilitate participation? Did the session take place at the prison? The audio recordings should be mentioned earlier, in the settings description.
  5. What level/type of education does “professionals” include? Later, health education or similar is mentioned, but this is not specified in the inclusion criteria or in the aims of the study. Who the “professionals” are is important for the interpretation of the results.
  6. Mix of reference styles is used, for instance sometimes Bardin is referred to as (12) and sometimes as Bardin (2016). Also, move the information on thematic analysis to the relevant paragraph (2.5)
  7. In the data analysis, a mix of deductive and inductive approaches are described, but it is not completely clear how they interact. Were the main categories pre-determined, or did they arise from the coding of the minimal fragment of content-units? This should be clarified in the methodology.
  8. Use past tense when referring to results.
  9. The methods describe the inductive development of sub-categories of themes, including the use of context such as facial expressions. This level is missing from the results. I.e. what are the different contextual challenges of the prison system, and how do the participants view them? Subcategories in results would give a better basis for the discussion. Currently, the (potential) sub-themes are presented for the first time in Discussion by referring to literature and not to results.
  10. Strengths and limitations under Discussion is a misleading heading. This section discusses strengths and limitations of the prison’s health-promoting activities, not the study, as would be expected. Please change this heading, and include a strengths and limitations section regarding the study design and methodology.

Author Response

Thank you for your careful and constructive review. We appreciate your thoughtful comments, which significantly contributed to improving the clarity, methodological transparency, and overall quality of our manuscript.

We have addressed all your suggestions in the revised version, including:

  • Removal of the reference from the abstract and substantial restructuring for clarity and balance;
  • Expansion of the results in the abstract and clearer distinction from the background;
  • Inclusion of a literature-based rationale for the chosen methods and sampling, and detailed explanation of the interview guide development;
  • Further detail regarding the data collection context, participant recruitment, and use of audio recordings;
  • Clarification of the professional profiles of participants and specification of inclusion criteria;
  • Expansion of the definition of "professionals" to include categories such as health care providers, social workers, and correctional officers;
  • Correction of reference inconsistencies and improved placement of methodological references;
  • Clarification of the interaction between deductive and inductive coding approaches;
  • Use of past tense throughout the results section and clearer integration of contextual data (e.g., non-verbal cues);
  • Inclusion of the use of context, such as facial expressions, for each of the categories described in the results. We are aware that, according to the literature, it is common for subcategories to emerge from the reported categories. However, in the case of the results analysed, we did not find sufficient robustness to interpret the categories that emerged from the discussion as a subcategory.
  • The previous ‘Strengths and limitations’ have been redefined and renamed ‘Institutional Opportunities and Constraints for Health Promotion’. A sub-topic was added at the end of the discussion on the limitations and strengths of the study and methodology.

We also revised the conclusion to include specific recommendations and implications, as suggested. Once again, thank you for your valuable feedback.

Reviewer 3 Report

Comments and Suggestions for Authors

The work developed by Machado et al. has its merits and after the following revisions it can be considered for publication in Nutrients:

The results mentioned in the abstract should be presented quantitatively. Some directions for further investigations could be pointed out.

The references are not formatted according to the journal’s guidelines.

The novelty of the present study is not clear. You must address this point in the Introduction. More data on the relation between eating habits in prisons among women and cardiovascular disease have to be given.

In the Material and Methods section, you need to explain how you established your sample size and its representativeness for the study population.

Table 2 has to be explained in the text.

You should include more studies carried out in other regions and compare their results with those obtained in your research. I miss further discussions from a worldwide perspective.

The Conclusions are adequate. However, as I previously mentioned, you should state some directions for future studies.

Author Response

Thank you for your thoughtful and constructive feedback. We are grateful for your recognition of the merits of our work and have addressed all your suggestions to improve the manuscript. Specifically:

  • The abstract was substantially revised. Results are now presented more clearly, structured under thematic categories and the conclusion highlights the implications and suggests directions for future research.
  • We removed the reference from the abstract and improved overall structure and readability.
  • References throughout the manuscript were reformatted to fully comply with the journal’s guidelines.
  • The introduction was rewritten and a new paragraph was added to better emphasize the novelty of the study — namely the focus on how professionals perceive barriers and strategies related to dietary habits and cardiovascular risk in women’s prisons — and to provide a more robust rationale supported by recent literature.
  • Additional updated and specific literature were added to clarify the connection between eating habits and cardiovascular disease in incarcerated women.
  • In the Methods section, we included a detailed explanation of the sampling strategy, its qualitative rationale, and how sample adequacy and representativeness were considered in this institutional context.
  • Table 2 is now explicitly introduced and explained in the results section. To improve readability, we relocated category descriptions to the main text and streamlined quotations.
  • The discussion now highlights the existing literature and articles written in a prison environment carried out in other regions, such as Brazil, were used.
  • Finally, the conclusion now includes concrete recommendations and clear directions for future research.

Thank you once again for your valuable input, which helped us strengthen the manuscript.

Reviewer 4 Report

Comments and Suggestions for Authors

Title:  Cardiovascular diseases in women's prisons: eating habits from 2 the perspective of professionals

---

@ This manuscript explores a topic that has received scant attention from the academic community yet is of crucial importance: the prevention of cardiovascular disease in women's prisons, with a particular focus on dietary habits. It provides insights of significant value from professionals working in this challenging environment.   The employment of a qualitative, exploratory design and focus group methodology is substantiated by robust research. The thematic content analysis employed, adhering to Bardin's protocol, is an appropriate method for capturing the nuances of participants' perspectives. The study provides a comprehensive description of the prison environment, including structural constraints and socio-cultural factors that influence eating habits. The incorporation of direct quotations and detailed tables serves to enhance the analytical depth of the study.

However, there are inconsistencies in the description of participant demographics. For instance, while the study focuses on women's prisons, the manuscript states that "two-thirds of the participants were married men," creating confusion regarding the composition of the professional demographic and the influence of gender on perspectives. The manuscript contains several long, complex sentences that reduce readability. Minor grammatical errors, typos and punctuation issues are also present. In particular, the sections titled "Results" and "Discussion" would benefit from clearer subheadings and smoother transitions. The flow of ideas is sometimes repetitive. While the focus group procedure is generally well described, additional details on the recruitment process, the criteria for achieving data saturation, and the steps in the coding process would strengthen the methodological transparency.

Notwithstanding the aforementioned criticisms, the manuscript makes a significant contribution to the extant literature on prison health, particularly by highlighting the interplay between dietary habits and cardiovascular risk among incarcerated women. Following revisions aimed at enhancing clarity, consistency, and detail in the methodology, this paper has the potential to make a significant impact in the domain of public health and prison healthcare policy.

@ Comments and suggestions by sections:

#Abstract

-The abstract provides a concise overview of the study's background, objectives, methods, and key findings. It is recommended that the sentences be rephrased in order to enhance clarity. For instance, the term "sociocultural and emotional factors" (Line 15) should be reviewed to ensure consistency in terminology.

-Furthermore, some sentences are overly lengthy; these should be broken down into shorter, more digestible parts.

-The research question or aim should be clearly stated in one sentence near the beginning.

# Introduction

-The introduction provides a concise overview of the global burden of cardiovascular diseases and situates the challenges within prison settings within the broader context.

-It is imperative to ensure that the introduction consistently reflects the focus on women's prisons.

-Furthermore, it is imperative to verify that all key claims are supported by appropriate citations. For instance, statistics on mortality rates and references to specific studies should be cross-checked for accuracy.

-Furthermore, transitions between paragraphs should be streamlined to guide the reader from global issues to the specific context of women's prisons.

-The manuscript provides background on the risk factors associated with incarceration, including sedentary lifestyles and dietary inadequacies.

-The discussion of specific risk factors should be integrated with literature on nutritional deficiencies and their impact on cardiovascular health.

-The use of overly technical language should be avoided wherever possible in order to ensure that the reader can understand the rationale behind the focus on dietary habits.

# Methods

-The description of the study design, participant selection, data collection and analysis is generally well articulated.

-However, further clarification is required regarding the demographics of the professionals involved. The reference to "married men" in the context of a women's prison (see the section on participant socio-demographics later in the text) requires further elucidation. It is unclear whether these individuals are healthcare providers or administrative staff.

-The text would benefit from further details on how the non-probabilistic sample was obtained and how data saturation was determined.

-The coding process merits further elaboration. While Bardin's protocol is cited, a brief description of how codes were derived and refined would enhance transparency.

-The ethical approval section is thorough; ensure that the study's limitations regarding confidentiality and data access are noted.

# Results

- The results are organised into thematic categories with supporting citations.

- Table 2 (Key Categories) is informative but could benefit from clearer formatting. Ensure that citations are concise and directly support the emerging themes.

- The manuscript should better articulate how each theme specifically relates to the overarching research question regarding dietary habits and CVD prevention.

- Address the inconsistency in the demographics of the participants. Clarify whether the sample consists only of professionals or whether there is direct involvement of inmates.

# Discussion

- The discussion section should thoroughly relate the findings to the existing literature, with particular focus on structural, socio-cultural and nutritional challenges.

- Organise the discussion using subheadings for each major theme (e.g. 'contextual challenges', 'socio-cultural barriers', 'intervention strategies').

- Some items will be repeated (e.g. challenges due to sedentary lifestyles and food quality issues). Consider summarising these points for brevity.

- While the need for improved policies is noted, adding specific recommendations could strengthen the impact.

- Ensure that the discussion clearly links back to the aim of the study and explains how the findings contribute to our understanding of CVD prevention in women's prisons.

# Conclusion

- The conclusion summarises the main findings and emphasises the need for interdisciplinary approaches.

-Strengthen the conclusion by explicitly stating the practical implications of the findings for prison health policy.

- Briefly mention the limitations of the study (e.g. small sample size, single-site study) and suggest directions for future research.

- Conclude with a clear call for more comprehensive studies and policy initiatives to improve nutritional conditions in prisons.

@ General language and style:

- There are minor grammatical errors and punctuation problems (e.g. unnecessary line breaks, inconsistent use of commas). Careful editing would improve overall readability.

- Ensure consistency in the formatting of headings, subheadings and references.

- Simplify overly complex sentences wherever possible. For example, break long sentences into shorter ones and ensure that each sentence conveys a main idea.

@ In summary, the manuscript provides a significant exploration of cardiovascular disease prevention in women's prisons, emphasising the role of diet and eating habits as experienced by professionals. While the study is methodologically sound and offers valuable insights, addressing inconsistencies in participant descriptions, streamlining the language, and reorganising sections for better clarity will strengthen the manuscript. The aforementioned recommendations, particularly those pertaining to methodological detail and the enhancement of thematic connections in the discussion section, are expected to contribute to the paper's meeting the standards for publication.

Comments on the Quality of English Language

Please, see report

Author Response

We sincerely thank you for the thorough and thoughtful review of our manuscript. Your detailed comments were instrumental in helping us improve the clarity, structure, and overall quality of our work.

In response to your suggestions:

  • The abstract was revised to improve clarity and flow. The research aim is now explicitly stated in a single sentence at the beginning, and overly long sentences were simplified for better readability. Terminology was reviewed to ensure consistency, including the term “sociocultural and emotional factors”.
  • The introduction was reworked to ensure a consistent focus on women's prisons. Transitions were improved, and citations supporting key claims (e.g., mortality statistics) were verified and updated. We also strengthened the link between incarceration-related risk factors and nutritional deficiencies in the context of cardiovascular disease.
  • Regarding the Methods section, we clarified the professional composition of participants and we considered that, since it was not our aim to establish a correlation between the sociodemographic variables gender and marital status of the participants (health professionals), we chose to describe as relevant to the study that four participants were married and two were single. We provided more detail on the sampling method, data saturation process, and coding procedures, including how codes were derived and refined based on Bardin’s protocol. No major changes were made to the “ethical considerations” section as it was already considered thorough; however, in accordance with the suggestions, we ensured that ethical safeguards regarding confidentiality and limited access to data were emphasized throughout the manuscript.
  • The Results section was reorganised for clarity. Table 2 was reformatted, and each thematic category is now better connected to the overarching research question. The text was revised to clearly state that the study involved only professionals, not incarcerated individuals.
  • The Discussion was significantly expanded and restructured with subheadings to guide the reader through each major theme. We reduced repetition, strengthened the connection to existing literature, and added specific recommendations for policy and practice. The contribution of the findings to the understanding of cardiovascular disease prevention in prisons is now more clearly articulated.
  • According to the suggestions of another reviewer, it made more sense to us to explicit practical implications for prison health policy and acknowledges study limitations (e.g., sample size and context) in the discussion. The conclusion now outlines directions for future research, including a call for more comprehensive studies and broader policy initiatives.

Additionally:

  • Language throughout the manuscript was simplified to improve readability. Long and complex sentences were revised, and minor grammatical issues and formatting inconsistencies were corrected.
  • The research team has contracted the English Editing service, which will carry out a more detailed revision of the language.
  • Headings and subheadings were standardised to improve organisation and coherence.

Once again, we thank you for your valuable feedback. We believe the manuscript is now considerably strengthened and aligned with the journal’s standards for publication.

Round 2

Reviewer 2 Report

Comments and Suggestions for Authors

The manuscript is now very clear in both the methods and results-presentation. 

Some final suggestions:

The paragraph on feelings towards the questions, ending with line 236 "The question took them by surprise; it was manifestly unexpected, leaving all individuals feeling quite uncomfortable.", is interesting, and the feelings of frustration in the professionals could perhaps be interpreted when relevant in the discussion/Or expanding the paragraph with more details on the uncomfortable questions.

For readability, the longer citations in the discussion could be exchanged with short parafrasing of the content, as the citations are found in table 2.

Author Response

We sincerely thank you for your positive feedback and final suggestions. We have carefully addressed all your comments in the revised version of the manuscript.
In the results section, we further explored the participants’ feelings of discomfort, adding explanations drawn from their own narratives (lines 239 – 255).
In the discussion, to improve the readability and clarity of the manuscript, we replaced the longer citations in the discussion with more concise paraphrased content, since the original excerpts are already presented in Table 2 (lines 400 – 448).
We hope these revisions respond effectively to your suggestions and contribute to a clearer, more coherent, and accessible manuscript. Thank you once again for your valuable input.

Reviewer 4 Report

Comments and Suggestions for Authors

Previous recommendations have been taken into account, and the manuscript has been significantly improved. It could be published in its current form.

Author Response

Thank you very much for your kind words and positive evaluation. We truly appreciate your feedback throughout the revision process and are glad the improvements were well received.